# Cloning and Characterization of Three Novel Enzymes Responsible for the Detoxification of Zearalenone

**DOI:** 10.3390/toxins14020082

**Published:** 2022-01-21

**Authors:** Yi Zhang, Xiaomeng Liu, Yunpeng Zhang, Xiaolin Zhang, He Huang

**Affiliations:** 1School of Pharmaceutical Sciences, Nanjing Tech University, Nanjing 211816, China; zhangyi_1012@126.com; 2Beijing Key Laboratory of Nutrition & Health and Food Safety, Nutrition and Health Research Institute, COFCO, Beijing 102209, China; liuxiaomeng1@cofco.com (X.L.); zyp_cau@163.com (Y.Z.)

**Keywords:** zearalenone, ZEN-degrading enzyme, ZHD101, protein expression

## Abstract

Zearalenone is a common mycotoxin contaminant in cereals that causes severe economic losses and serious risks to health of human and animals. Many strategies have been devised to degrade ZEN and keep food safe. The hydrolase ZHD101 from *Clonostachys rosea,* which catalyzes the hydrolytic degradation of ZEN, has been studied widely. In the current research, three new enzymes that have the capacity to detoxify ZEN were identified, namely CLA, EXO, and TRI, showing 61%, 63%, and 97% amino acids identities with ZHD101, respectively. Three coding genes was expressed as heterologous in *Escherichia coli* BL21. Through biochemical analysis, the purified recombinant CLA, EXO, TRI, and ZHD101 exhibited high activities of degrading ZEN with the specific activity of 114.8 U/mg, 459.0 U/mg, 239.8 U/mg, and 242.8 U/mg. The optimal temperatures of CLA, EXO, TRI, and ZHD101 were 40 °C, 40 °C, 40 °C, and 45 °C, and their optimum pH were 7.0, 9.0, 9.5, and 9.0, respectively. Our study demonstrated that the novel enzymes CLA, EXO, and TRI possessed high ability to degrade ZEN from the model solutions and could be the promising candidates for ZEN detoxification in practical application.

## 1. Introduction

Zearalenone (ZEN) is a non-steroidal estrogenic mycotoxin biosynthesized by *Fusarium* strains, which was firstly isolated from moldy corn in 1962 [1] and widely contaminating musty corn, wheat, and other grains [2]. Since its structure is similar to estrogen, excessive intake of ZEN can cause anabolic activity and hyper-estrogenism in the reproductive organs of animals [3,4]; in addition, ZEN has also been known to show hepatotoxicity, hematotoxicity, immunotoxicity, and genotoxicity potential [5,6]. High incidences of ZEN was reported from different countries [7], including Europe, Asia, and Africa [8,9,10,11], resulting in huge loss of agriculture commodities [12]. Hence, many methods have been explored to reduce ZEN contamination [13].

Physical and chemical methods, including intensive heating, irradiation, adsorption, alkaline hydrolysis, and ozonation, were widely used to detoxify toxin-contaminated feedstocks [14,15,16,17]. However, ZEN is highly thermostable, insensitive to UV irradiation, and is not easy to destroy during feed processing [18,19]. As for chemical methods, only strong alkalis or oxidants could break its chemical structure [20]. Although these methods achieved reduction in ZEN concentration, there were many disadvantages, such as non-selectivity, destruction of nutrients, and introduction of pollution in the feeds [21,22]. In comparison, microbial strains [23,24,25,26] or isolated enzymes [27,28,29] are always used as the biological methods to degrade ZEN to non-estrogenic derivatives, among which ZEN-degrading enzymes were a more attractive alternative due to their safety, efficiency, and irreversibility. It was reported that ZEN-degrading enzyme ZHD101 isolated from *Clonostachys rosea* cleaved to the lactone ring of ZEN and yielded non-toxic alkylresorcinol product [30,31,32,33]. The encoding gene *zhd101* has been expressed in different hosts successfully. In 2002, Takahashi-Ando et al. expressed *zhd101* gene in *Escherichia coli* heterologously [30]. Then, the *zhd101* gene was expressed in *E. coli* and *Saccharomyces cerevisiae* fused with an enhanced green fluorescence protein (EGFP) to visually monitor its detoxification activity [31]. The recombinant *E. coli* completely degraded 2 μg/mL ZEN in 1 h, while the recombinant *S. cerevisiae* only degraded 75% of ZEN in the medium in 4 d. In 2005, Higa-Nishiyama et al. introduced the *zhd101* gene into a model monocotyledon rice plant, and the protein extract from leaves showed the significant ability of degrading ZEN [32]. In addition, the *zhd101* gene was expressed in *Pichia pastoris* GS115, and it only took 30 min to degrade 10 μg/mL ZEN by the purified enzyme [34]. In 2017, the *zhd101* gene was introduced into *Lactobacillus reuteri* Pg4, and the transformed strain *L. reuteri* pNZ-*zhd101* successfully expressed ZHD101 and acquired the capacity to degrade ZEN [35]. Until now, ZHD101 is still mostly studied for its high ZEN-degrading ability and mechanism.

*E. coli* is a commonly used host strain for heterologous protein expression considering its several prominent features, including fast cell growth in minimal media, low acetate production when grown in high levels of glucose, low protease abundance, and an amenability to high-density culture [36]. Extensive research has been investigated on expression of several highly homologous protein in different strains, including ZHD518 [37], ZENG [38], ZHD795 [39], and ZEN-jjm [40], which presented 65%, 99%, 62%, and 99% amino acid identities with ZHD101. All these existing ZEN-degrading lactonases possessed a relatively low optimum temperature and thermostability, which limits its use in the feed industry. Therefore, it is extremely important to excavate new genes encoding ZEN-degrading enzymes to further improve the degradation condition of ZEN.

In this study, three potential ZEN-degrading enzymes were screened based on the k protein sequences of reported ZHD101 in the NCBI, and the genes were expressed in *E. coli.* Results showed the purified enzymes could degrade ZEN efficiently, which will provide resources for the removal of ZEN in feed industry.

## 2. Results

### 2.1. Selection of the ZEN-Degrading Enzymes

To search for resources, the amino acid sequence of ZHD101 was compared with the database reserved in GenBank by protein BLAST. Indeed, 29 stains were identified (Figure 1), among which three proteins (GenBank accession number: XP_016613277.1, XP_013255140.1, AHG29544.1) shared 61.22%, 62.88%, and 97.35% identities with ZHD101 (GenBank accession number: ALI16790.1), namely CLA, EXO, and TRI, respectively (Figure 2).

The sequences of the *cla*, *exo*, *tri,* and *zhd101* gene from the *Cladophialophora bantiana*, *Exophiala aquamarina*, *Trichoderma aggressivum*, and *C. rosea* contained complete open reading frame (ORF). The length of sequences were 795 bp, 792 bp, 792 bp, and 792 bp, respectively. The overall G+C content of the genes were 41%, 42%, 42%, and 46%.

### 2.2. Expression and Purification of the Four Proteins

The genes encoding CLA, EXO, TRI, and ZHD101 proteins were cloned into expression vector pET28a (+) and confirmed successful construction by restriction endonuclease of *Bam*HI and *Hin*dIII (Figure 3) and sequence analysis.

SDS-PAGE analysis revealed that the purified CLA, EXO, TRI, and ZHD101 migrated as single band with a molecular mass of about 29 kDa (Figure 4), which was consistent with the theoretical molecular mass of 29.08 kDa, 29.3 kDa, 28.89 kDa, and 28.75 kDa, respectively.

### 2.3. Enzymatic Properties of the Four ZEN-Degrading Enzymes

CLA showed high levels of activity at pH 7.0 to 7.5 (>90%), while EXO, TRI, and ZHD101 showed high levels of degrading activity at pH 9.0 to 9.5 (>90%) (Figure 5). The optimal pH for activity of four ZEN degrading-enzymes, including CLA, EXO, TRI, and ZHD101, was at pH 7.0, 9.0, 9.5, and 9.0, respectively. 

CLA, EXO, TRI, and ZHD101 showed optimal temperature degrading ZEN at 40 °C, 40 °C, 40 °C, and 45 °C, and all of them performed more than 50% of the enzyme activity in the temperature range from 20 °C to 45 °C. However, when temperature increased above 50 °C, the relative activity decreased significantly (Figure 6A–D).

CLA, EXO, TRI, and ZHD101 were stable and retained at least 90% of activity after incubation at 40 °C for 10 min. After being incubated at 50 °C for 5 min, TRI and ZHD101 showed more than 50% of the maximum activity, but the activities substantially reduced above 50 °C (Figure 6E).

## 3. Discussion

Numerous physical and chemical methods have been widely studied to prevent the growth of mycotoxigenic fungi and to detoxify ZEN-contaminated agricultural products over the past years [41]. However, most of them were not safe and useful in practice [42]. Biological decontamination exhibited an attractive alternative for minimizing the negative effects of mycotoxins because of its mild reaction conditions, high efficiency, and environmentally friendly nature [43]. The microorganisms degrading ZEN mainly involved two pathways, including the adsorption of the ZEN onto the walls of the microbial cells and degradation of the ZEN caused by biotransformation [44,45].

ZEN-degrading enzymes are able to convert ZEN into non-toxic or low-toxic products, and ZEN detoxification by ZHDCP and carboxypetidase enzyme produce H-ZEN ((E)-2,4-dihydroxy-6-(10-hydroxy-6-oxo-1-undecen-1-yl)-benzoic acid) and D-ZEN ((E)-1-(3, 5-dihydroxy-phenyl)-10-hydroxy-1-undecen-6-one), which were recognized as less estrogenic derivatives [46]. Viksoe-Nielsen and Soerensen discovered that the zearalenone in a feed product can be degraded into non-toxic substances by treating the feed product with laccases from *Myceliophthora thermophila*, *Polyporus pinsitus,* and *Streptomyces coelicolor* [47]. Effective detoxification can be achieved by deliberate introduction of purified enzymes during feed processing [48,49]. At present, the research of ZEN-degrading enzyme has become a hot spot, and it might be the significantly potential detoxification means for the future industrial application [44,48]. Lactono-hydrolase, such as ZHD101, ZHD518, ZENG, and ZENC, have been studied widely, and their capabilities to degrade ZEN and chemical structures have been reported [31,37,38,50]. 

Here, the amino acid sequence of ZHD101 was compared by protein BLAST in the GenBank database, and three hypothetical proteins, CLA, EXO, and TRI, sharing 61.22%, 62.88%, and 97.35% homologies with ZHD101, were chosen and expressed in *E. coli*. The recombinant strains could remove ZEN from the model solutions rapidly and efficiently.

The biochemical analysis showed that the optimal temperature and pH of CLA, EXO, TRI, and ZHD101 were 40 °C, 40 °C, 40 °C, and 45 °C and 7.0, 9.0, 9.5, and 9.0, respectively. It was found that the pH of the two potential ZEN-degrading enzymes EXO and TRI was identical with the ZHD of *C. rosea* IFO 7063 [30] and most other lactonases, which showed an alkaline optimal pH of generally 8.5 to 10.0 [28,30,31,34], while the protein CLA exhibited the maximum activity at pH 7.0 in 50 mM Na_2_HPO_4_-citric acid buffer and decreased rapidly under alkaline conditions. This was significantly different from most of zearalenone lactono-hydrolases except for three proteins, namely ZHD518 [37], ZENC, and ZENG [38], whose optimal pH was 8.0, 8.0, and 7.0, respectively. 

CLA had more than 85% enzyme activity (relative to maximum activity) at 30–40 °C and EXO, TRI, and ZHD101 remained over 75%. The four ZEN-degrading enzymes showed a great amount loss of enzymatic activity beyond 65 °C. It was interesting that four ZEN-degrading enzymes still have more than 50% activity within 10 min of incubation even at 20 °C, which is significant for industrial ZEN removal. Previous reports suggests that most of the ZEN-degrading lactonases, including RmZHD from *Rhinocladiella mackenziei* [28], ZHD101 from *C. rosea* [31], ZHD518 from *R. mackenziei* [37], ZENC from *Neurospora crassa* [50], and ZENG from *G. roseum* [38], displayed the maximum activities at a relatively low temperature ranging from 37 °C to 45 °C. In addition, it was noticed that the CLA was homologous as expressed by Hui et al. [51]. According to their research, the same protein named CbZHD was expressed in *E. coli*, and CbZHD showed maximum activity at pH 8.0 and 35 °C, which is different from our studies. It was speculated that the different reaction buffers might result in the different optimal conditions.

The present results demonstrated that CLA, EXO, TRI, and ZHD101 were promising enzymes that could be used for removing ZEN in the feed industries. It is noteworthy that the CLA even possess a higher enzyme activity than ZHD101 in a neutral pH condition. For industrial applications, large-scale production of the efficient ZEN-degrading enzyme and a more cost-effective expression system were urgent to be developed and employed.

## 4. Conclusions

In conclusion, three novel ZEN-degrading enzymes named CLA, EXO, and TRI were cloned and produced using a heterologous *E. coli* expression system. It showed the optimal temperature and pH of CLA, EXO, and TRI were 40 °C, 40 °C, and 40 °C and 7.0, 9.0, and 9.5, respectively. The recombinant enzymes exhibited high detoxification of ZEN and could be used as potential candidates for commercial applications.

## 5. Materials and Methods

### 5.1. Plasmids, Strains, Chemicals, and Medium

*E. coli* JM109 was used as the cloning host strain. The plasmid pET28a (+) and *E. coli* BL21 (DE3) were used for gene expression. Luria–Bertani (LB) medium was prepared to culture the strains. Restriction enzymes (*Ba*
*m*HI and *Hin*dIII), Ex Taq DNA polymerase, T4 DNA ligase, and other related enzymes were obtained from Takara (Dalian, China). Isopropyl-β-D-1-thiogalactopyranosid (IPTG) and kanamycin were purchased from Solarbio (Beijing, China). Zearalenone was bought from Pribolab (Qingdao, China) and dissolved in acetonitrile as a standard stock solution (1 mg/mL). Acetonitrile and methanol of HPLC grade were purchased from ANPEL laboratory Technologies (Shanghai, China).

### 5.2. Cloning of the Four ZEN-Degrading Enzyme Genes and Expression in E. coli

The sequences of ZEN-degrading enzyme encoding genes *cla*, *exo*, *tri*, and *zhd101* were synthesized with addition of *Bam*HI and *Hin*dIII restriction sites at both ends by Tsingke Company (Nanjing, China), then amplified by PCR with specific Primers (Table 1). The obtained PCR product was cloned into pET-28a (+) between the *Bam*HI and *Hin*dIII restriction sites and fused in-frame with the α-factor signal peptide. The recombinant plasmids were then transformed into *E. coli* BL21 (DE3) through electroporation for expression. Subsequently, the recombinant *E. coli* BL21 (DE3) strains were cultivated in 250 mL LB medium containing 50 µg/mL kanamycin at 37 °C until OD_600nm_ reached 0.6–0.8, induced with 0.2 mM IPTG shaken at 20 °C for 6 h. The cells were collected by centrifugation (6500 rpm, 5 min) and stored at −80 °C before purification.

### 5.3. Enzyme Purification and Molecular Mass Determination

The cells pellets were resuspended in the lysis buffer (50 mM Tris-HCl, 0.5 M NaCl, pH 7.5) and ultrasonically disrupted on ice for 15 min (ultrasonic 3 s, pause 6 s).The lysates were centrifuged at 4 °C, 12,000 rpm for 15 min to remove cell debris. The supernatants were loaded onto the Ni-NTA column equilibrated with 50 mM Tris-HCl buffer (pH 8.0) and eluted with a gradient concentration of imidazole solution (20–500 mM), then dialyzed in a buffer of 200 mM NaCl and 25 mM Tris-HCl, pH 7.5. The molecular mass of proteins was analyzed by 12% SDS-PAGE. The concentration of protein was determined with BCA Protein Assay Kit (Solarbio, China).

### 5.4. Enzymatic Substrate Degrading Activity

The enzyme activity was characterized by substrate depletion. Each assay solution (500 µL) contained 10 µL substrate (1 mg/mL ZEN in acetonitrile), 10 µL diluted enzyme solution, and 480 µL buffer. After incubation at the optimum temperature for 10 min, 500 µL methanol was added to terminate the reaction. After filtration, the sample was analyzed using a Shimadzu LC-20AT HPLC system equipped with a Shimadzu RF-20A fluorescence detector. Aliquots of the tested solutions (10 µL) were applied on a Agilent Eclipse XDB-C18 column (4.6 mm × 150 mm, 5 µm) and eluted with a acetonitrile-water-methanol mix (46:46:8) at a rate of 1.0 mL/min. The column was kept at 40 °C. The excitation and emission wavelengths were 274 nm and 440 nm, respectively. Concentrations of ZEN were determined based on retention times and peak areas compared to ZEN standard stock solution. The peak area vs. ZEN concentration standard curve was plotted and was in very good correlation. One unit of ZEN-degrading enzyme activity was defined as the amount of enzyme required to degrade 1 µg substrate per minute under standard assay conditions.

### 5.5. Effects of Temperature and pH on the Enzyme Activity and Stability

To investigate the influence of pH on the four ZEN-degrading enzymes, enzymatic activity was determined in Glycine-HCl (50 mM, pH 2.0), Na_2_HPO_4_-Citric acid (50 mM, pH 3.0–7.5), Tris-HCl (50 mM, pH 7.5–9.0), and Glycine-NaOH (50 mM, pH 9.0–11.0). The optimal temperature for activity of four ZEN-degrading enzymes was measured at 20 °C, 30 °C, 37 °C, 40 °C, 45 °C, 50 °C, and 65 °C at optimal pH. The thermo-stability was assayed by pre-incubating the enzyme for 2 min, 5 min, 7 min, and 10 min at 40 °C, 50 °C, and 65 °C, and then, the residual enzymatic activity was determined under optimal conditions. The untreated enzyme was used as the control.

## Figures and Tables

**Figure 1 toxins-14-00082-f001:**
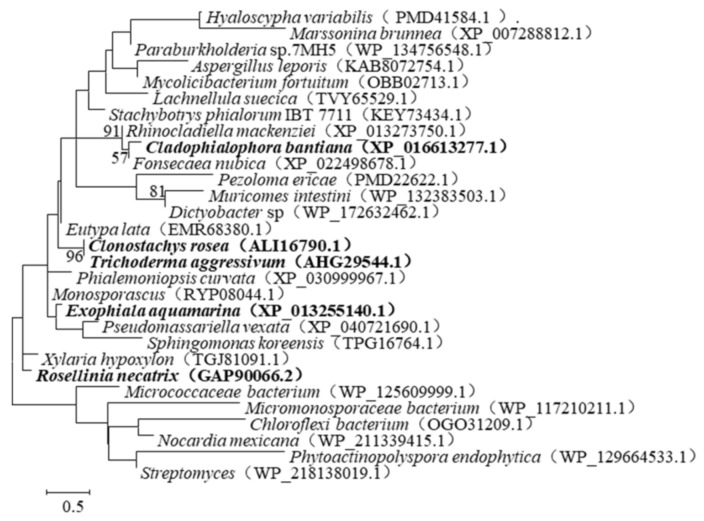
A phylogenetic tree was constructed based on the amino acid sequences of ZHD101 by means of neighbor-joining analysis. Bootstrap values (*n* = 1000 replicates) were reported as percentages.

**Figure 2 toxins-14-00082-f002:**
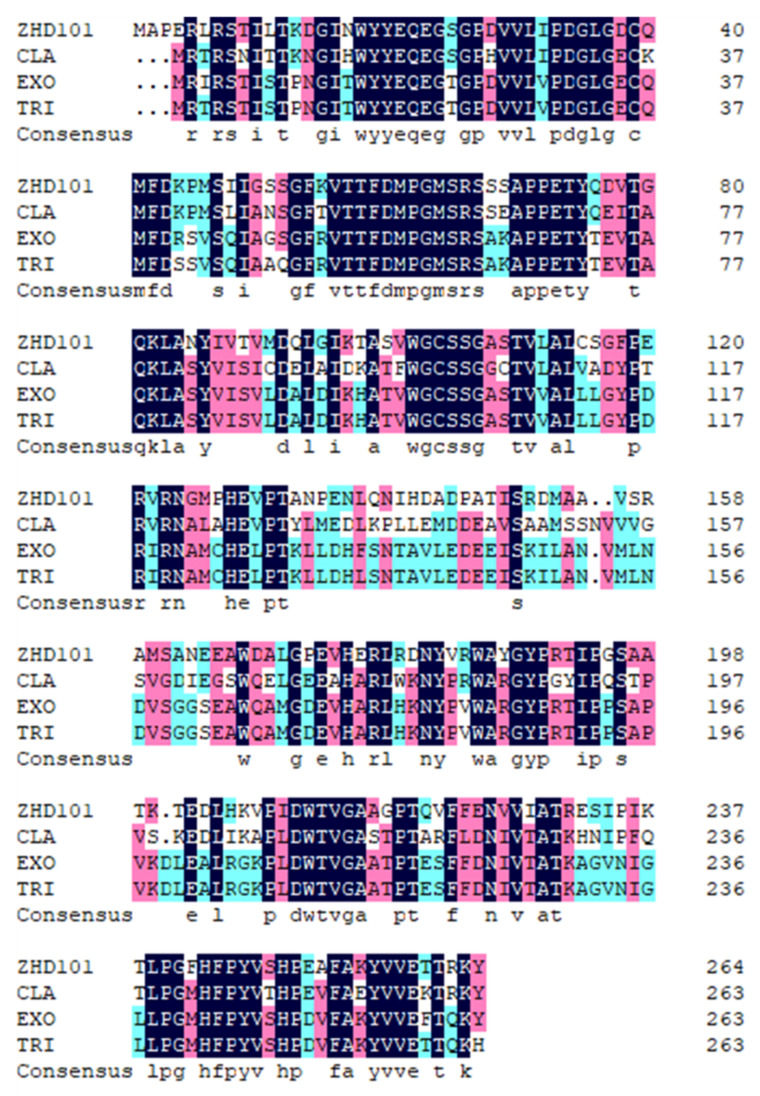
Amino acid sequence alignment of ZEN-degrading enzymes.

**Figure 3 toxins-14-00082-f003:**
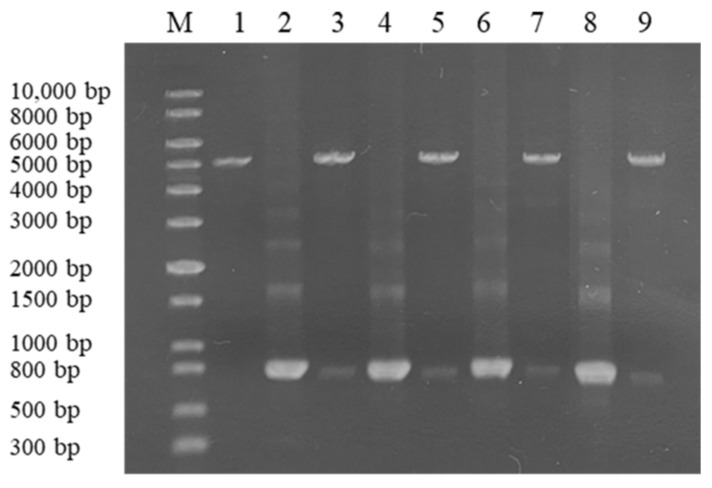
Electrophoresis analysis of expression plasmid. Lane M: 1 kb plus DNA marker; lane 1, pET28a (+); lane 2, 4, 6, 8, *cla*, *exo*, *tri*, *zhd101*; lane 3, 5, 7, 9, recombined plasmid pET28a (+)-*cla*, pET28a (+)-*exo*, pET28a (+)-*tri*, and pET28a (+)-*zhd101* were digested by *Bam*HI and *Hin*dIII.

**Figure 4 toxins-14-00082-f004:**
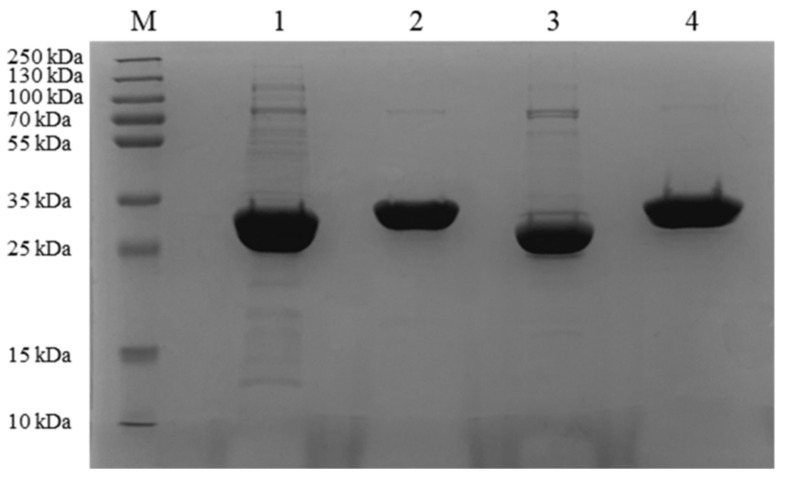
SDS-PAGE analysis of the recombinant protein. Lane M, prestained protein ladder (250, 130, 100, 70, 55, 35, 25, 15, and 10 kDa); lane 1, purified recombinant CLA; lane 2, purified recombinant EXO; lane 3, purified recombinant TRI; lane 4, purified recombinant ZHD101.

**Figure 5 toxins-14-00082-f005:**
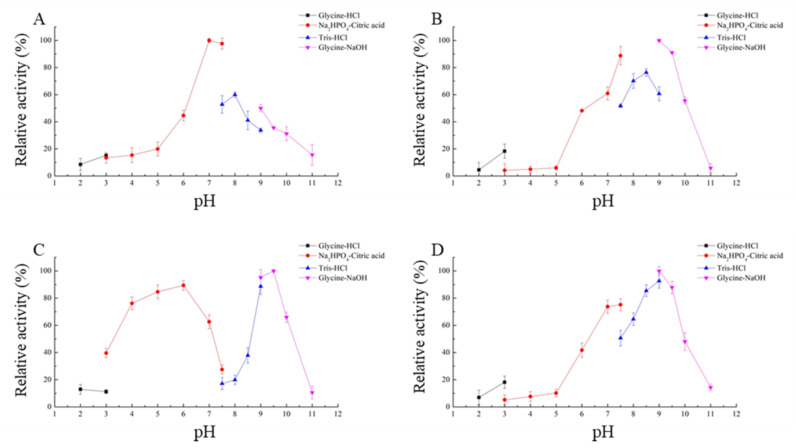
Effects of different pH on the enzyme activity of recombinant ZEN-degrading enzymes. (**A**) Effects of pH on CLA catalytic activity; (**B**) effects of pH on EXO catalytic activity; (**C**) effects of pH on TRI catalytic activity; (**D**) effects of pH on ZHD101 catalytic activity.

**Figure 6 toxins-14-00082-f006:**
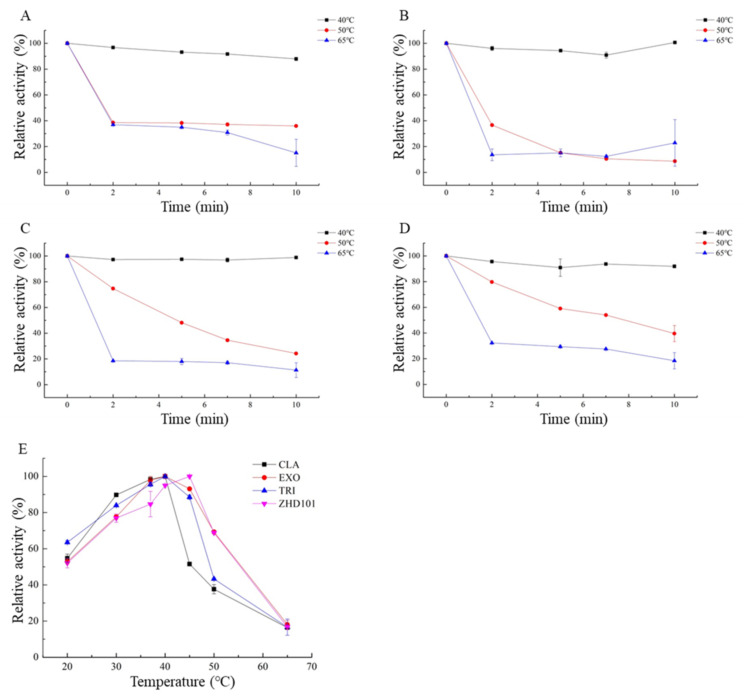
Effects of different temperature on the enzyme activity and stability of recombinant ZEN-degrading enzymes. (**A**) Effects of temperature on CLA catalytic activity; (**B**) effects of temperature on EXO catalytic activity; (**C**) effects of temperature on TRI catalytic activity; (**D**) effects of temperature on ZHD101 catalytic activity; (**E**) thermostability of recombinant ZEN-degrading enzymes.

**Table 1 toxins-14-00082-t001:** Primers used in this study.

Primer Name	Amplicon Gene	Sequences (5′-3′)	Restriction Site
CLA-F	*cla*	ATAGGATCCATGGCTCCAGAAAGATTGAG	*Bam*HI
CLA-R		CGCAAGCTTTCACAAGTACTTTCTAGTAGTTT	*Hin*dIII
EXO-F	*exo*	ATAGGATCCATGAGAACCAGATCCAACAT	*Bam*HI
EXO-R		CGCAAGCTTTCACAAGTACTTTCTAGTTTTTT	*Hin*dIII
TRI-F	*tri*	TCAGGATCCATGAGAATCAGATCCACCAT	*Bam*HI
TRI-R		GGCAAGCTTTCACAAGTACTTCTGAGTAAACT	*Hin*dIII
ZHD101-F	*zhd101*	ATAGGATCCATGAGAACGCGGAGCACGAT	*Bam*HI
ZHD101-R		CGCAAGCTTCTACAGATGTTTCTGCGTCGTTT	*Hin*dIII

## Data Availability

The data that support the findings of this study are available from the corresponding author on reasonable request.

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
