# Peer review of "Cloning and Characterization of Three Novel Enzymes Responsible for the Detoxification of Zearalenone"

_toxins, 2022, doi:10.3390/toxins14020082_

Round 1

Reviewer 1 Report

Dear Authors,

I reviewed the submitted version, however, the idea of investigation seems interesting, there are lots of works in improving the article.

I indicated some of them in the attached file 

Author Response

Thank you very much for your comments, which are valuable for improving the quality of our manuscript. 

The title was changed to "Cloning and Characterization of Three Novel Enzymes Responsible for the Detoxification of Zearalenone" and others were modified in the revised manuscript according to your kind suggestions by using blue text. .

Reviewer 2 Report

Thank you very much for the oportunity to review this article.

The authors evaluated the Heterologous Expression and Characterization of Three Novel Zearalenone-Degrading Enzymes.

The manuscript is well organized amd well written.

Some minor remarks are following.

Please enrich the Introduction section so that in total it has at least 40 references

Line 6. strageies...you mean strategies?

Line 67. datebase...you mean database?

Please rewrite the conclusions with the main findings of your research.

Author Response

Thank you very much for your approval. We have revised the manuscript according to your suggestions.

1. Please enrich the Introduction section so that in total it has at least 40 references

Response: The more references were added in the introduction section in the revised manuscript.

2. Line 6. strageies...you mean strategies?

Response: Yes, it meant strategies, and it was corrected in the revised manuscript.

3. Line 67. datebase...you mean database?

Response: Yes, it meant database, and it was corrected in the revised manuscript.

4. Please rewrite the conclusions with the main findings of your research.

Response: The conclusions were rewritten in the revised manuscript.

Reviewer 3 Report

The manuscript reported the degradation ability of newly identified enzymes – CLA, EXO, and TRI on zearalenone suggesting their possible application in feed industry as zearalenone detoxifier. While the paper is of great value, I strongly suggest that the authors improve the manuscript both in content and language especially the discussion and conclusion sections.

Comments

Line 7 - 8, correct sentence. Change “…Clonostachys rosea catalyzes…” to “…Clonostachys rosea which catalyzes…”. Delete the comma after ZEN

Line 11-13, not clear, correct accordingly

Line 15, change “removing” to “degrading”

Line 27, change “resulting in the huge loss in agriculture” to “resulting in huge loss of agriculture commodities”

Line 28, change “reduce the contamination caused by ZEN” to “reduce ZEN contamination”.

This sentence should be moved to the first paragraph. And the second paragraph start with the sentence “Physical …

Line 29, delete “the”

Line 30, change “…although these methods could achieve the effect of ZEN removal…” to “…although these methods achieved reduction in ZEN concentration…”

Line 31, change “…destroying nutrients…” to “…destruction of nutrients…”

Line 33, delete “a”

Line 34-36, correct the sentence “It was reported that a ZEN-degrading enzyme ZHD101 isolated from Clonostachys rosea 34 was studied extensively [11-14], and it could cleave the lactone ring of ZEN and yield 35 non-toxic alkylresorcinol product [11].” to “It was reported that ZEN-degrading enzyme ZHD101 isolated from Clonostachys rosea cleaved to the lactone ring of ZEN and yielded non-toxic alkylresorcinol product [11-14].

Line 44, resolving?

Line 44-45, correct sentence

Line 59-61, correct sentence

Line 122, change “had” to “have”. Correct subsequently in the manuscript

Line 127, change “removing” to “degrading”

Line 128, change “containing” to “including”

Line 130-134, correct sentences

Line 147-149, three lactonases? Correct sentence

Line 152-154, confusing sentence. The paragraph (Line 152-172) is poorly written. Correct accordingly

Line 165, ZHD518, include the source.

Author Response

We appreciate your kind suggestions on the earlier version of our manuscript, which are valuable for improving the quality of our manuscript. We have revised the manuscript according to the suggestions by using blue text.

Round 2

Reviewer 1 Report

Dear Authors,

Thanks for your efforts to cover my comments. The MS is seems ready to get published.

Author Response

Thank you very much for your good comments and hard work.

Reviewer 3 Report

I strongly advise that the authors seek the services of a native English speaker to proofread and edit the manuscript. The authors should correct the sentences below.

Line 5-6, Zearalenone…  

Line 9, change "had" to "have"

Line 30-31, Hence….

Line 138-140, ZEN-degrading… mention example of non-toxic or low-toxic product

Line 140-142, As a matter…

Line 162, Literature from previous studies? Either use literature or previous studies. Correct accordingly

Line 169-171, it can be …

Author Response

Dear Reviewers,

Thank you very much for your comments, which are valuable for improving the quality of our manuscript. We have revised the manuscript according to the suggestions.

Line 5-6, Zearalenone…  

Response: The sentence was modified in the revised manuscript.

Line 9, change "had" to "have"

Response: It was changed in the revised manuscript.

Line 30-31, Hence….

Response: The sentence was modified in the revised manuscript.

Line 138-140, ZEN-degrading… mention example of non-toxic or low-toxic product

Response: The example was added in the revised manuscript.

Line 140-142, As a matter…

Response: The sentence was modified in the revised manuscript.

Line 162, Literature from previous studies? Either use literature or previous studies. Correct accordingly

Response: The sentence was modified in the revised manuscript.

Line 169-171, it can be …

Response: The sentence was modified in the revised manuscript.